# Prevalence of Obesity and Associated Risk Factors among Children and Adolescents in the Eastern Cape Province

**DOI:** 10.3390/ijerph19052946

**Published:** 2022-03-03

**Authors:** Sibusiso Cyprian Nomatshila, Sikhumbuzo A. Mabunda, Thandi Puoane, Teke R. Apalata

**Affiliations:** 1Department of Public Health, Walter Sisulu University, Mthatha 5117, South Africa; 2George Institute for Global Health, University of New South Wales, Sydney 2033, Australia; smabunda@georgeinstitute.org.au; 3School of Population Health, University of New South Wales, Sydney 2033, Australia; 4Department of Public Health, University of the Western Cape, Cape Town 7535, South Africa; tpuoane@gmail.com; 5Department of Laboratory Medicine, Walter Sisulu University, Mthatha 5117, South Africa; tapalata@wsu.ac.za

**Keywords:** overweight, obesity, children, non-communicable diseases, risk factors

## Abstract

Obesity is a global public health concern that begins in childhood and is on the rise among people aged 18 and up, with substantial health consequences that offer socioeconomic challenges at all levels, from households to governments. Obesity and associated risk factors were investigated in children and adolescents in the Eastern Cape Province of South Africa. A cross-sectional study was conducted at Mt Frere among 209 conveniently selected participants using anthropometric measurements and a structured questionnaire. Chi-squared statistics or Fisher’s exact test were used to evaluate the risk factors predicting different outcomes such as hypertension or diabetes mellitus. A 5% level of significance was used for statistical significance (*p*-value 0.05). The prevalence of overweight or obesity among females when using waist circumference (2.7%), triceps skinfold (6.9%), and body mass index cut-offs (16.4%) were respectively higher when compared to those of males. About 89% engaged in physical activities. After school, 53% watched television. About 24.9% of participants did not eat breakfast. Most of overweight or obese participants (92.9%) brought pocket money to school. Use of single anthropometric measurements for assessing nutritional status indicated inconclusive results. Strengthening parental care, motivation for consumption of breakfast and limiting pocket money for children going to school are important steps to improve child health.

## 1. Introduction

Childhood overweight and obesity are reported to exist among children and adolescents and continue through into adulthood and are associated with the early development of non-communicable diseases [1]. Global obesity has been reported to have increased by threefold since 1975, with the overweight prevalence among people aged 18 years and older recorded at 1.9 billion, of which more than 650 million were obese [2]. This increasing trend was reported by the non-communicable disease (NCD) Risk Factor Collaboration, which confirmed that globally, there had been an increase in the obesity prevalence from 3.2% in 1975 to 10.8% in 2014 among men, whilst it increased from 6.4% to 14.9% among women [3]. This rate was 11% for men and 15% of women in 2016 [2].

These increasing trends were observed in similar ways for both high income (HIC) and low- and middle-income countries (LMIC). For instance, in Mexico, as a middle-income country, a prevalence study on obesity among school-aged children reported the prevalence of obesity to have increased from 8.9% in 1999 to 14.6% in 2012 among children and adolescents [4]. Over a twenty-year period between 1991–2011 in Slovenia, as a high income country, tendencies showed that obesity odds were growing at a faster pace than overweight per year, especially among male children and adolescents [5]. Within a period of six years, the prevalence of overweight increased exponentially among boys from 1.5% in 2009 to 6.5% in 2015, while the same trends were observed among girls, where overweight increased from 1.0% to 6.1% in China, which is categorized as an upper-middle-income country [6]. Similarly, in the upper- middle-income countries like China, identical trend was also observed in a Chinese study in which the combined prevalence of overweight and obesity among children displayed an increase over a period of time [7].

Obesity has increased in low- and middle-income countries, not just among economically productive, middle-aged, and rich individuals, but also among children and adolescent populations of low socioeconomic status (LMICs). Of late, Africa has been seen to be experiencing high rates of overweight, with an estimated escalation in the prevalence of childhood overweight from 4% in 1990 to 7% in 2011 [8]. This increase was in line with an increase that was observed in a 2010 South African study, which revealed that obesity prevalence among White children had increased by 7.0%, from 20.3% in the 2010 assessment to 27.3% in 2013 [9]. This increase was considered to be double that of Black children, which only showed an increase of 3.0% (10.3% to 13.3%) in the same period of assessment [9]. Obesity prevalence rates in Sub-Saharan African countries classified as LMICs ranged from 3.3% to 18.0%, according to the World Health Organization (WHO), making obesity a substantial risk factor for diabetes mellitus and cardiovascular diseases in Africa’s modernized areas [10]. According to statistics, South Africa has one of the highest rates of child obesity in Africa [11,12].

Obesity and its associated risk factors are likely to continue to afflict an increasing number of people in low- and middle-income countries [13]. Around 6% of deaths worldwide, mainly in LMICs, can be attributable to risk factors such as little or no physical activity [14]. Urgent attention was needed to halt the increasing rates of childhood obesity [15,16]. There is a correlation between overweight and obesity in school-aged children, which has been linked to an increased risk of obesity later in life [17]. Childhood overweight and obesity is a critical public health concern that could have serious health repercussions as well as impose a socioeconomic burden on families and governments [18].

The use of indirect measures of obesity (such as BMI, waist circumference, and waist-to-hip ratio) as the sole means of determining obesity has an impact on the credibility of some of these findings [18]. While the South African public health system focuses on undernutrition and disregards other types of nutritional status as non-threats, it is also vital to distinguish between the contributions of fat and muscle weight to body weight in order to reduce NCD risk factors. [19]. Given that a lower mean BMI does not necessarily rule out the presence of obesity [13], it is critical to develop adequate protocols and measurements for decisive and accurate diagnosis of overweight or obesity whilst risk factors are being understood and addressed. Other metrics, such as skinfold, would be required for a convincing diagnosis of obesity [13]. This study aimed at investigating prevalence of obesity and associated risk factors among children and adolescents.

## 2. Materials and Methods

### 2.1. Study Design

A cross-sectional study targeting children aged between 9–18 years was conducted among PURE (Prospective Urban Rural Epidemiology) study households in KwaBhaca, Eastern Cape Province of South Africa. The PURE is a multi-country longitudinal study which seeks to determine the relative contributions of societal influences, diet, built environment and lifestyle behaviors on obesity and chronic health NCDs [20].

### 2.2. Setting

The study was conducted in KwaBhaca in the Eastern Cape province of South Africa. KwaBhaca is a rural town with a population of over 5000 people, and the unemployment rate is estimated to be over 76%, with the majority of residents living below the poverty line and earning between 1001 and 2500 Rands (67 to 168 US Dollars) per month [21].

### 2.3. Population

One racially homogenous community and all homes that made up the PURE study participants were formerly targeted for recruitment for the current investigation. Only children aged 9 to 18 who were members of PURE research households and lived with them were eligible to participate, while those who were visiting were not. Of the 110 families contacted, 102 agreed to have their children participate in the study. This population of interest was selected because the PURE studies never focused on children but only on the adult population.

### 2.4. Sampling Procedure

Because there were no street names, a cluster sampling of dwellings in chosen villages under the clan chiefs was carried out. Eligible members of the selected households were recruited as cohort study participants. For the PURE project [20], sampling was planned to get a widely representative sample of the population of adults aged 35 to 70 years in each household. As a result, convenience sampling was employed to find 209 individuals aged 9 to 18 years old.

### 2.5. Data Collection

Participants’ risk variables known to be associated with obesity were assessed using an English-language structured, researcher-administered questionnaire. There were both open-ended and closed-ended questions. To guarantee that the questionnaire was appropriately translated, it was first translated into isiXhosa and then back into English. All researchers were trained on the use of the questionnaire and instruments. The questionnaire obtained information on socio-demographic variables like age, education, location, and employment status as well as lifestyle factors like physical activity, diet, tobacco smoking, and alcohol use. Blood pressure, blood glucose and anthropometric measures like weight, height, mid-upper arm circumference, waist circumference and triceps skinfold were measured with specific instruments as described below. The questionnaire had a Cronbach’s alpha 6-item scale reliability co-efficient of 71.3%.

The weight in kilograms (kg) was measured with an Ironman^TM^ BC-554 electronic weighing scale (Tanita Corporation, Tokyo, Japan). The weighing scale was placed on a firm, flat, and consistent surface. Participants were allowed to wear only light clothes like vests or tee-shirts, and they were encouraged to take off excessive clothing and shoes. They were made to stand upright and still during weighing; they stood in the center of the scale with their feet slightly apart and remained still till the scale recorded their weight.

A Health-o-meter^®^ 402KLROD (Health O Meter, Illinois, USA) was used to measure the height of participants. To measure and record the height of a participant, the stadiometer was placed on even ground. Shoes and hair accessories were removed. Participants stood on the flat floor with their feet slightly apart. The back of the head, shoulder blades, buttocks, calves, and heels all touched the vertical board. The legs were kept straight and the feet flat, with the heels and calves touching the vertical board of the scale. The participant’s head was positioned so that a horizontal line from the ear canal to the lower border of the eye socket ran parallel to the base of a flat surface. To keep the head in this position, the bridge between the researcher’s thumb and forefinger was held over the child’s chin. Still keeping the head in this position, the headpiece was pulled down to rest firmly on top of the head and compress the hair. The measurement was read and recorded as the child’s height in centimeters to the last completed 0.1 cm (1 mm).

The FHI360 MUAC (FHI 360, Washington, USA) calibrated tape was used to measure MUAC for the participants. Participants stood erect with their arms relaxed. The sleeve of the left arm (unless contraindicated) was rolled up to expose the arm. The midpoint of the upper left arm between the acromion process on the shoulder blade and the tip of the olecranon process of the ulna was determined. This was done by finding the top of the shoulder and the tip of the elbow using tape. The tape was held firmly on the two points. While holding it still, the part on the tip of the elbow was taken and the tape folded by putting the two points together and marking the mid-arm. While the arm was left hanging loosely, the palm facing inwards, the tape was wrapped gently but firmly around the arm at mid-arm.

A calibrated Seca 201 Body Fitness waist to hip ratio ergonomic tape (Seca, Hamburg, Germany) was used to measure the abdominal circumference. Measurements were taken directly on the participants’ abdomens without any form of clothing. Participants were asked to hold their arms around as if one was giving him/herself a hug. The right ilium of the pelvis in the hip area and the uppermost lateral border of the ilium were located. The measuring tape was extended around the waist and was made to sit parallel to the floor and lay snug but was not made to compress the skin. The measurement was recorded to the nearest 0.1 cm.

Slim Guide^®^ skinfold calipers (Ningbo Finer Medical Instruments Co. Limited, Zhejiang, China) were used to measure skin thickness for each participant. A vertical fold was located on the posterior midline of the upper arm. Measurements were taken halfway between the acromion (bony point of the shoulder) and olecranon processes (bony point of the elbow). The arm was held freely at the side of the body while the measurements were taken.

A standardized protocol adapted for the PURE study [20] was used to record the anthropometric measurements. All cut-offs for MUAC, waist circumference, triceps skinfold and BMI were based on WHO and set at +1 SD for overweight and +2 SD for obesity [22].

Overweight was defined as +1 SD (equivalent to the 85th percentile) in the WHO reference, with a cut-off of 25 (kg/m^2^), whereas obesity was defined as +2 SD (equivalent to the 97th centile), with a cut-off of 30 (kg/m^2^). For MUAC, waist circumference and triceps skinfold, overweight and obesity were defined as the 85th percentile in the WHO reference.

Nutritional value for food items eaten by participants was collected through a seven-day recall. Standard measurements for food items were in accordance with South African Food Data System (SAFOODS) [23].

### 2.6. Data Analysis

Data were collected and coded in Microsoft Excel 2013 (Microsoft Corporation, Seattle, USA) before being exported to Stata 14.1 (STATA Corp LP, College Station, Texas, USA) for analysis. Presence and range check settings were used to prevent double entries by subject number and to minimize erroneous field entries.

The distribution of numerical variables was explored using the Shapiro-Wilk test, histogram, and/or box-and-whisker plot. Numerical data that are not normally distributed are reported using the median and interquartile range (IQR).

The variance ratio test was used to test the equality of variances before the means and variances from two different populations were compared. The relevant two-sample *t*-test, or the Satterthwaite’s Modified *t*-test, or the Wilcoxon Sum rank test (Mann–Whitney test), was used depending on the equality of variances and as to whether the variables were normally distributed or not. The Kruskal–Wallis test was used to compare three weight categories and the demographic variables.

Categorical variables are presented using frequency tables, percentages and graphs. Two or more categorical variables were compared using contingency tables (e.g., 2 × 2 Table) and the expected frequencies were calculated to determine the type of test to use for the purpose of determining the extent of the relative associations. If the expected frequencies were ≥ 5 then the Chi-squared test (Chi^2^) was used, and if the expected frequencies were < 5 then the Fisher’s exact test was used.

The level of significance was set at 5% (*p*-value ≤ 0.05) for statistical significance. Nutritional value was analyzed based estimates provided by the SAFOODS database.

### 2.7. Ethical and Legal Considerations

Ethical approval was granted by the Faculty of Health Sciences Human Research Ethics and Biosafety committee of Walter Sisulu University with ethical clearance number 070/15. Permission to conduct the study was further granted by the Eastern Cape Department of Health, whilst parents of children assented and the children consented to participating in the study. Principles of autonomy, beneficence, non-maleficence and justice were implemented in line with the provisions of Helsinki declaration.

### 2.8. Validity and Reliability

The questions for this standardized and validated questionnaire were adapted from the PURE research baseline questionnaires [24], South African National Health and Nutrition Examination Survey (SANHANNES) [25] and Food Frequency Questionnaire [26] to match the study’s needs. To check reliability, relevance of research questions, phrasing, question format, and ability to measure what was intended, the instrument for data collection was based on questions validated through a small-scale pilot of five people not participating in the core study. A test-and-re-test approach was used to confirm the accuracy of the adjusted questions after the pilot study.

## 3. Results

### 3.1. Demographic Charecteristics

Table 1 shows that there were more female participants (*n* = 116/209) in the study than males (*n* = 93/209). Of all participants, 13.40% (28/209) were classified as being overweight or obese. Overweight or obese participants were older (median = 13.95 years) than normal (med = 12.2) years and underweight (med = 13.0) years participants, and this was not statistically significant (*p* = 0.081). Though it was not statistically significant (*p* = 0.138), among the total number of overweight or obese participants, 67.9% (19/28) were female. Most overweight or obese participants (46.4% or *n* = 13/28) were aged 12–15 years. The majority of the participants who were overweight or obese (53.6%; *p* = 0.085) were in primary school. All three (3/209, or 2.0%) participants who were not in school were classified as normal. Reasons for not being at school included mental retardation (*n* = 2) and one person who had not been enrolled at school for poorly defined reasons.

Even though not statistically significant, a combined 57.1% (16/28) of participants who were classified as being overweight or obese were cared for by people receiving either a child support grant or an old age grant. This was not statistically different (*p* = 0.979) from comparable normal or underweight participants (59.06% or 107/181). The majority (92.9% or *n* = 26) of those classified as being overweight or obese reported carrying pocket money to school. As such, only 30 out of 209 respondents (14.4%) did not carry pocket money to school.

Watching television was reported as a common activity among participants, with 75% (or *n* = 21) of overweight or obese participants reporting having watched television after school, and it was statistically significant (*p* = 0.013). Though it was not statistically significant (*p* = 0.299), skipping breakfast was reported by 33.33% of the overweight or obese participants, and 24.5% (36/147) of the normal and 21.9% (7/32) underweight participants.

### 3.2. Anthropometric Measurements

Table 2 presents the anthropometric measures which shows that there was no statistical difference when comparing the obesity prevalence of males and females when categorized by MUAC (*p* = 0.256); waist circumference (*p* = 0.253); triceps skinfold (*p* = 0.352) and BMI (*p* = 0.221). In total, 33/209 (15.8%) of the participant were malnourished. However, females (median = 11 cm) had larger triceps skinfolds than their male counter parts (8cm) and this difference was statistically significant (*p* < 0.0001).

Table 3 confirms moderate agreement between all anthropometric measurements used to test overweight or obesity.

### 3.3. Physical Activity

Table 1 further shows that even though 80.8% (138/146) of normal and 30/33 (90.9%) underweight participants walked to school, this was not statistically different (*p* = 0.216) from what was reported by three quarters (21/28 or 75.0%) of the overweight or obese participants. Similarly, there was no statistical difference in the reported median walking time (20 min) between the two groups of participants (*p* = 0.171). With most sports activities reported to take place at school, those whose weight was assessed as being normal were more likely to have participated in sport (134/146 or 91.1%) than the 78.6% (22/28) of those who were overweight or obese (*p* = 0.153). Although not significant (*p* = 0.076), most participants who were classified as being normal reported playing soccer (72/146, or 43.9%) and netball (27/146, or 16.5%). Furthermore, normal participants were statistically more likely to play sports daily (46.9% or 60/146; *p* = 0.0492) and not watch television (TV) after school (49.3% or 73/146; *p* = 0.013).

Participants were further asked to list all activities they engaged in after school and could enter as many activities as they wanted. On average, only 454 activity lists were entered, and this on average represents almost two activities per participant. Sport activities and house chores had the most entries with 38.1% or *n* = 173 and 35.3% or *n* = 160, respectively. Almost a quarter (24.7% or *n* = 112) of the entries referred to watching television (TV) after school. Sleeping and playing video games were reported in 3 (0.7%) and 6 (1.3%) entries, respectively.

### 3.4. Dietary Habits of South African Rural Children

During breakfast, carbohydrates are generally flour-based (bread), maize-based (soft porridge), cereal (refined and/or unrefined), poorly defined (previous day’s food) or a combination of any of the groups. Additionally shown in Table 2 is that most participants who were classified as being overweight or obese also 33.3% (9/27) had soft porridge, followed by 22.2% (*n* = 6) who had cornflakes and weet-bix or wheat-based cereal (*n* = 2 or 7.4%) during breakfast, but this was not statistically significant (*p* = 0.698). There were, however, 52 individuals who generally did not eat breakfast at all, and these individuals respectively comprised a third (9/27) of those classified as being overweight or obese and 24.0% (43/179) of those classified as being normal or underweight. Table 3 shows agreement between anthropometric measurements collected from studied population.

The nutritional value of foods reported as eaten by children during breakfast and the nutrient adequacy ratio by gender for breakfast foods eaten by children is presented in Appendix A and Appendix A, respectively.

#### 3.4.1. School Meals

Even though 197/206 participants (95.6%) reported the presence of school feeding schemes in their schools, 60.7% (125/206) did not have tuckshops or food vendors in their schools, and 21.84% (45/206) brought their own lunch to school. Furthermore, 84.4% (152/180) of those classified as being of normal weight or underweight were also given pocket money when going to school, compared to 92.9% (26/28) of those classified as being overweight or obese, but this was not statistically significant (*p* = 0.385) (Table 2). Of the 362 reported frequencies of purchases from the school tuckshop, chips, sweets, and fat cakes (dough fried in oil) accounted for 81.5% (295/362) of all the purchases. Viewed differently, chips, sweets, and fat cakes were favourite purchases for 118/209 (56.5%), 104/209 (49.8%), and 73 (34.8%) participants, respectively (Figure 1). As the fourth most purchased item (10.8% of all purchases), polony (processed meat sausage) was the favourite purchase for 39 participants (18.6%). Table 4 indicate foods eaten by children provide as lunch by the school in terms of national school nutrition program. The nutritional value for foods eaten during lunch served at school and nutrient adequacy ratio by gender for foods eaten during lunch served at school as part of nutrition program are presented in Appendix A and Appendix A, respectively.

#### 3.4.2. After School Meal and Dinner

A total of one hundred and fifty-seven (157/206 or 76.2%) of participants always ate a meal immediately after school, whilst twelve (5.71%) of participants reported that they never had a meal immediately after school. This meal precedes dinner and follows the lunch meal they had at school. Table 5 show the foods that are reported to be eaten by the participants after school and Table 6 indicates foods eaten by participants during dinner. Appendix A shows the nutritional adequacy ratios of the meals that participants had after school. The nutritional values of respective foods that are reported to be eaten by children after school and during dinner, are presented in Appendix A and Appendix A.

## 4. Discussion

Overweight and obesity have been proven to exist in this study among children in rural communities of South Africa. This is in line with the study conducted in Israel where it was found that 30.6% of the population were identified as overweight or obese in 2015–2016 [27]. A rate of 13.2% overweight and 6.8% obesity were also identified among school going children in Nepal in 2018 [28]. However, this study discovered that the use of a single method for the diagnosis of overweight or obesity was inconclusive. Despite their extensive use, Body Mass Index and Waist Circumference have major limits for use as obesity screening measures in clinical practice [29]. In a study conducted among university students, it transpired that different outcomes and obesity prevalence were obtained when different anthropometric measures were used [30]. This was further supported by Amirabdollahian and Haghighatdoost [31].

This study identified association between social support system and prevalence of obesity among children. This was also discovered in a 2015 study conducted across Organisation for Economic Cooperation and Development countries, which discovered that the prevalence of childhood obesity was negatively connected to social spending on children [32]. This is further supported by another study which concluded that increased government spending is likely to exacerbate childhood obesity [33].

Children bringing pocket money to school were found to have a high association to unhealthy food consumption, which leads to obesity, which is a risk factor for noncommunicable diseases [34]. Another study concluded that healthy body weight was associated to having less pocket money [35]. Another study conducted in China in 2015 found that certain child and parental characteristics were linked to children’s pocket money, which increased the chance of bad eating habits and being overweight or obese [36].

The current study discovered that three quarters of overweight child participants had sedentary lifestyle like watching television after school. It was also observed in another study that there was a significant link between childhood obesity and computer use, as well as television viewing [37]. This was also determined in another study which discovered that obesity was prevalent among children and adolescents who are exposed to screen media [38]. Furthermore, NCD risk factors such as overweight or obesity included sitting for more than 2 h per week while listening to music or the radio [39]. Excessive television viewing was linked to the consumption of sugary snacks by children, which is a risk factor for the development of non-communicable disease [40].

Skipping of breakfast is one of the significant risk factors for the development of NCDs. Th study identified that skipping breakfast was reported by a third of the overweight and obese participants. The rising prevalence of skipping breakfast has been linked to an increase in noncommunicable diseases [41]. Among German schoolchildren, substantial correlations between skipping breakfast and having a body fat percentage at or above the 95th percentile were discovered [42]. Breakfast skipping was also linked to obesity in Brazilian adolescents [43]. Taking breakfast daily was found to be lowering chances of the development of obesity among children [44]. This is consistent with other findings that found a positive association between having breakfast and physical activity among school-going children [45]. Breakfast of poor quality is a risk factor for the development of overweight or obesity, as well as chronic conditions that can lead to adulthood obesity, hypertension, and diabetes [46,47].

This study observed that more than three-quarters of those who were overweight or obese did not participate in sporting activities. In 2012, physical exercise was found to be negatively related with the risk of childhood obesity [48]. Although there are many factors that contribute to the development of obesity in children, one of the most important drivers of overweight and obesity is a decrease in metabolic rate [49]. In comparison to overweight and obese Saudi Arabian adolescents, normal weight participants reported the highest levels of physical activity [50]. Another study found that children aged 9–11 had a higher frequency of obesity and a lower percentage of physical activity [51]. Obesity in children has been found in studies to reduce the number of friends a child has, resulting in less valuable time spent on physical activity with friends [52]. Obesity has a wide-ranging negative psychological influence, with the unfortunate result that a child may avoid company and prefer to be alone rather than with friends [53,54,55]. Altman and Wilfley echoed these thoughts when they stated that fat children and adolescents are frequently subjected to severe stress and prejudice in all sectors, including schools and personal interactions [56]. Another study indicated that children who ate breakfast on a daily basis had a higher positive school climate connectedness and better academic achievements than those who skipped breakfast or ate it seldom [57]. However, a family’s socioeconomic status plays an important role in the availability of food in the household including the consumption and type of breakfast [57].

This study further identified that overweight and obese participants had maize-based soft porridge and cereals in the morning for breakfast. In a study done in Lesotho, the majority of participants took maize porridge (56.1%) on a regular basis, and added sugar to their food (74.2%), and more than 27% of the girl population of the study was overweight or obese [58]. This risk is further exacerbated by purchasing of unhealthy snacks and meals by children at school as was revealed by this study. Though this study revealed that meals taken by children at school, after school, and during dinner provide substantial nutrient value, children were still exposed to purchasing chips, sugary drinks, sweets, and other unhealthy foods at schools. Findings of this study are in line with other studies which found that South African schoolchildren are sold unhealthy food options, such as low-nutrient energy-dense foods (e.g., chips, candies) and sugar-sweetened beverages [59]. Govender advocated for a ban on the sale of fizzy drinks and junk food in school canteens, citing major worries about the rising number of overweight and obese school children [60]. This is a risk factor not only for children’s development but also for obesity and NCDs as children did not spend the energy that they had taken in [61]. This excess energy is then stored in the form of fat in the body, thus increasing the body weight and exposing a child to obesity, cardiovascular diseases, and diabetes [62].

## 5. Limitations

Only children from households that took part in the prospective urban-rural epidemiology study (PURE) in Mt. Frere were studied. The general populace of Mt. Frere was not covered. As a result, this study cannot be extrapolated to the full Mt. Frere children’s population, thereby restricting the study’s scope and sample size. Furthermore, the resultant children with excess body weight could be a skewed estimate of the true effect in the population. However, this is a reflection of what was happening within the population of interest. Following that, this study’s contribution to the literature on obesity levels and associated risk factors in rural settings, as well as establishing mitigating frameworks for preventing and managing NCD risk factors, remained.

## 6. Conclusions

This study set to identify obesity and NCD risk factors among children of families who participated in a previous multi-country NCD study (PURE study). Participants were a combination of a small proportion of obese children and those with malnutrition, poor dietary habits (non-balanced diet, skipping meals, etc.), non-healthy meal options at school and a sedentary lifestyle that is somehow weakened by the fact that many participants walked to school. These are risk factors that need to be addressed using a life-course approach to the prevention of NCDs. Strategies should incorporate an enabling policy framework in schools and a curriculum which places an emphasis on prevention of future health risks.

## Figures and Tables

**Figure 1 ijerph-19-02946-f001:**
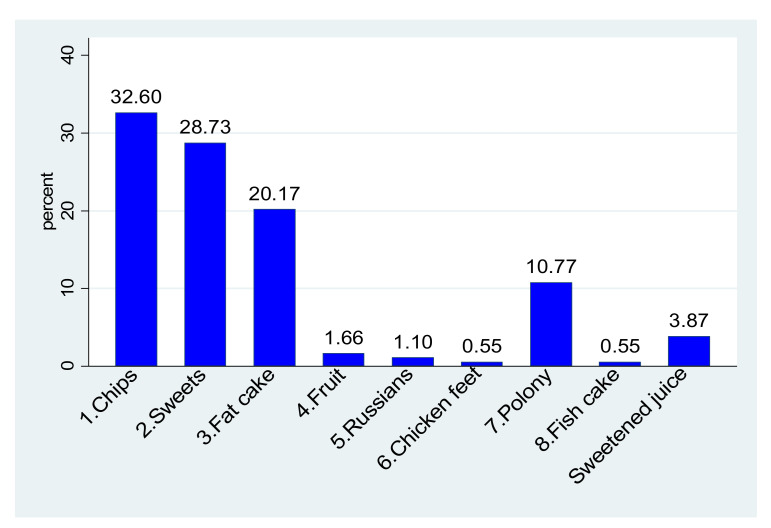
Items commonly purchased from school vendors and/or tuckshop.

**Table 1 ijerph-19-02946-t001:** Demographic characteristics and lifestyle of children and adolescents.

Demographic Characteristics (*n* = 209)	Overweight or Obese		Normal		Underweight		*p*-Value
Weight status; *n* (%)	28	(13.4)	148	(70.8)	33	(15.8)	-
Age, years; med (IQR **)	13.95	(4.1)	12.2	(4.4)	13.0	(3)	0.081 ^♣^
Sex; *n* (%)							
Males	9	(32.1)	65	(43.9)	19	(57.6)	0.138
Females	19	(67.9)	83	(56.1)	14	(42.4)	
Age, years; *n* (%)							
≤11	7	(25.0)	64	(43.2)	10	(30.3)	0.089
12–15	13	(46.4)	53	(35.8)	19	(57.6)	
>15	8	(28.6)	31	(21.0)	4	(12.1)	
Current grade at school; *n* (%)							
None	0	(0.0)	3	(2.0)	0	(0.0)	0.085 *
Primary school	15	(53.6)	102	(68.9)	28	(84.9)	
High school	13	(46.4)	43	(29.1)	5	(15.2)	
^~^ Mode used to get to school; *n* (%)							
Walk	21	(75.0)	118	(80.8)	30	(90.9)	0.216
Public, private or school transport	7	(25.0)	28	(19.2)	3	(9.1)	
Average walking time to school, minutes; med (IQR)	20	(15)	20	(20)	15	(20)	0.171 ^♣^
^^^ Participation in sports; *n* (%)							
Yes	22	(78.6)	134	(91.1)	30	(90.9)	0.153
No	6	(21.4)	13	(8.8)	3	(9.1)	
^$^ Where most sport takes place; *n* (%)							
School	9	(40.9)	69	(52.3)	11	(35.5)	0.171
Home	5	(22.7)	15	(11.4)	8	(25.8)	
School and home	8	(36.4)	48	(36.4)	12	(38.7)	
^$^ Specific sport; *n* (%)							
Soccer	7	(33.3)	72	(43.9)	17	(54.8)	0.074 **
Netball	7	(33.3)	27	(16.5)	5	(16.1)	
Cricket	0	(0.0)	12	(9.0)	1	(3.2)	
Rugby	1	(4.8)	0	(0.00)	0	(0.0)	
Hoopla hoops	0	(0.0)	0	(0.0)	1	(3.2)	
Other	6	(28.6)	44	(33.1)	7	(22.6)	
^#^ Frequency of sport; *n* (%)							
Daily	7	(31.8)	60	(46.9)	17	(60.7)	0.092 *
Weekly	11	(50.0)	34	(26.6)	4	(14.3)	
Occasionally	4	(18.2)	34	(26.6)	7	(25.0)	
Play sport after school; *n* (%)							
Yes	17	(60.7)	98	(66.2)	22	(66.7)	0.563
No	11	(39.3)	50	(33.8)	11	(33.3)	
Watch TV after school; *n* (%)							
Yes	21	(75.0)	75	(50.7)	15	(45.5)	0.013
No	7	(25.0)	73	(49.3)	18	(54.5)	
Play TV or Video games; *n* (%)							
Yes	2	(7.1)	4	(2.7)	0	(0.0)	0.185 *
No	26	(92.9)	144	(97.3)	33	(100.0)	
Sleep; *n* (%)							
Yes	0	(0.0)	2	(1.4)	1	(3.0)	1.00 *
No	28	(100.0)	146	(98.6)	32	(97.0)	
House chores; *n* (%)							
Yes	23	(82.1)	114	(75.7)	23	(69.7)	0.453
No	5	(17.9)	34	(24.3)	10	(30.3)	
Traditional games; *n* (%)							
Yes	9	(32.1)	40	(27.0)	11	(33.3)	0.666
No	19	(67.9)	108	(73.0)	22	(66.7)	
~ Taking breakfast; *n* (%)							0.299
Yes	18	(66.7)	111	(75.5)	25	(78.1)	
No	9	(33.3)	36	(24.5)	7	(21.9)	
^~^ Breakfast; *n* (%)							
Cornflakes	6	(22.2)	28	(19.1)	3	(9.4)	0.698
Weet-bix/wheat-base	2	(7.4)	11	(7.5)	1	(3.1)	
Sifted maize	9	(33.3)	65	(44.2)	17	(53.1)	
Previous day’s food	1	(3.7)	7	(4.8)	4	(12.5)	
None	9	(33.3)	36	(24.5)	7	(21.9)	
^¥^ Bring own lunch to school; *n* (%)							
Yes	6	(21.4)	33	(22.6)	6	(18.2)	1.000
No	22	(78.6)	113	(77.4)	27	(81.8)	
^^^ Lunch core starch; *n* (%)							
Maize based	26	(92.9)	129	(87.8)	30	(90.9)	0.342
None	1	(3.6)	15	(10.2)	3	(9.1)	
Both	1	(3.6)	3	(2.0)	0	(0.0)	
Starch dinner; *n* (%)							
Maize based	21	(75.0)	108	(73.0)	23	(69.7)	0.142 *
Rice	6	(21.4)	40	(27.0)	10	(30.3)	
None	1	(3.6)	0	(0.0)	0	(0.0)	
Caregiver employment status; *n* (%)							
Yes	2	(7.1)	11	(7.4)	2	(6.1)	1.000 *
No	26	(92.9)	137	(92.6)	31	(93.9)	
Caregiver income source; *n* (%)							
None	4	(14.23)	22	(14.9)	3	(9.1)	0.979 *
Disability grant	3	(10.7)	5	(4.4)	3	(9.1)	
Old age grant	8	(28.6)	43	(29.1)	11	(33.3)	
Child support grant	8	(28.6)	45	(30.4)	8	(24.2)	
Support from spouse	2	(7.1)	12	(6.8)	2	(6.1)	
Support from parent	0	(0.0)	3	(2.0)	0	(0.0)	
Support from other children	1	(3.6)	5	(3.4)	2	(6.1)	
Vending	0	(0.0)	0	(0.0)	1	(3.0)	
Pension or retirement	0	(0.0)	3	(2.0)	0	(0.0)	
Foster care grant	0	(0.0)	1	(0.7)	1	(3.0)	
Employment	2	(7.1)	11	(7.4)	2	(6.1)	
^^^ Pocket money to school; *n* (%)							
Yes	26	(92.9)	122	(83.0)	30	(90.9)	0.560
No	2	(7.1)	25	(17.0)	3	(9.1)	

^ *n* = 208; ^¥^
*n* = 207; ^$^
*n* = 185; ^#^
*n* = 178; ~ *n* = 206; * Fisher’s exact test was used; ** IQR = Interquartile Range = 75th percentile minus (-) 25th percentile; med = median; ^♣^ Kruskal–Wallis test was used.

**Table 2 ijerph-19-02946-t002:** Anthropometric measurements for children.

Characteristics (*n* = 209)	Male		Female		*p*-Value
	*n* = 93		*n* = 116		
MUAC; *n* (%)					
Obese	0	(0.0)	3	(2.6)	0.256 *
Not obese	93	(100.0)	113	(97.4)	
Waist circumference, cm ^#^; med (IQR **)	62	(10)	63	(13)	0.677
^~^ Waist circumference; *n* (%)					
Obese	0	(0.0)	3	(2.7)	0.253
Not obese	93	(100.0)	110	(97.4)	
Triceps skinfold, cm ^#^; med (IQR **)	8	(4)	11	(7)	<0.0001
Triceps skinfold; *n* (%)					
Obese	3	(3.2)	8	(6.9)	0.352
Not obese	90	(96.8)	108	(93.1)	
BMI, kg/m^2^; med (IQR **)	17.42	(3.9)	18.41	(4.6)	0.088
BMI; *n* (%)					
Obese	1	(1.1)	4	(3.5)	0.221
Overweight	8	(8.6)	15	(12.9)	
Not obese	65	(69.9)	83	(71.6)	
Malnourished	19	(20.4)	14	(12.1)	

~ *n* = 206; # cm = Centimeters; ** IQR = Interquartile Range = 75th percentile minus (-) 25th percentile; * Fisher’s exact test was used; med = median; ~ Waist circumference could not be measured on three participants as one refused and two were wheelchair bound; NB: All cut-offs were set based on WHO [27].

**Table 3 ijerph-19-02946-t003:** Agreement between anthropometric measurements.

Anthropometric Measure	Agreement (%)	Kappa	*p*-Value
BMI	100	1	1
Waist Circumference	87.9	0.1718	<0.0001
Triceps Skinfold	90.0	0.4175	<0.0001
MUAC	88.0	0.1720	<0.0001

**Table 4 ijerph-19-02946-t004:** Foods eaten by children and adolescents during lunch served at school.

	Type of Food	*n* (%)
Carbohydrates	Maize	186 (88.6)
	Maize and rice	4 (1.9)
Bread	Brown bread	22 (10.5)
	White bread	22 (10.5)
Vegetables	Beans/legumes	176 (83.8)
	Potatoes	44 (21.0)
	Cabbage	147 (70)
	Carrots	102 (48.6)
	Beetroot	64 (30.5)
	Spinach	8 (3.8)
	Mixed vegetables	2 (1.0)
	None	23 (11.0)
Fruit	Apple	112 (53.3)
	Orange	3 (1.4)
	Plum	1 (0.5)
	None	93 (44.3)
Meat	Red meat	5 (2.4)
	Chicken/pork	49 (23.3)
	Processed meat (sausage/vienna/polony)	16 (7.6)
	Eggs/Fish	45 (21.4)
	Soup	72 (34.3)
Dairy products	Fermented milk (Amasi)	11 (5.2)
Usual drink at school	Water	172 (81.9)
	Fruit juice	11 (5.2)
	Sweetened juice	20 (9.5)
	Fizzy drink	1 (0.5)
	Other forms of beverages	1 (0.48; 0.07–3.4)
	None	5 (2.38; 0.99–5.6)

**Table 5 ijerph-19-02946-t005:** Foods eaten by children and adolescents at home after school.

	Type of Food	*n* (%)
Carbohydrates	Maize	89 (42.4)
	Rice	73 (34.8)
	Maize and rice	1 (0.5)
Bread	Brown bread	83 (39.5)
	White bread	43 (20.5)
Vegetables	Beans/legumes	38 (18.1)
	Potatoes	48 (22.9)
	Cabbage	51 (24.3)
	Carrots	14 (6.7)
	Beetroot	2 (1.0)
	Spinach	17 (8.1)
	Mixed vegetables	3 (1.4)
	None	96 (45.7)
Fruit	Apple	1 (0.5)
	Banana	1 (0.5)
	None	208 (99.1)
Meat	Red meat	3 (1.4)
	Chicken/pork	62 (29.5)
	Processed meat (sausage/vienna/polony)	17 (8.1)
	Eggs/Fish	21 (10)
	Soup	67 (31.9)
Dairy products	Fermented milk (Amasi)	4 (1.9)

**Table 6 ijerph-19-02946-t006:** Foods eaten by children and adolescents during dinner.

	Types of Food	*n* (%)
Carbohydrates	Maize	153 (72.9)
	Rice	56 (26.7)
Bread	Brown bread	8 (3.9)
	White bread	39 (18.6)
Vegetables	Beans/legumes	87 (41.4)
	Potatoes	81 (38.6)
	Cabbage	111 (52.9)
	Carrots	33 (15.7)
	Beetroot	3 (1.4)
	Spinach	22 (10.5)
	Mixed vegetables	13 (6.2)
	None	26 (12.4)
Fruit	Apple	1 (0.5)
	Banana	1 (0.5)
	None	208 (99.1)
Meat	Red meat	13 (6.2)
	Chicken/pork	133 (63.3)
	Eggs/Fish	12 (5.7)
	Soup	35 (16.7)
Dairy products	Fermented milk (Amasi)	15 (7.1)

## Data Availability

Data are available on request though it will be guided by research regulations, Protection of Personal Information Act, and the confidentiality agreement with participants.

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
