# Peer review of "Prevalence of Obesity and Associated Risk Factors among Children and Adolescents in the Eastern Cape Province"

_ijerph, 2022, doi:10.3390/ijerph19052946_

Round 1

Reviewer 1 Report

Comments;
The study group - 209 children aged 8-19 is too small to calculate the prevalence of overweight and obesity.
There are no data on the number of white and black children in the study group.
The subgroup of 28 children with excess body weight is too small to be statistically valid.
It is worth noting that among 209 children, malnutrition is still a bigger problem - it concerns 33 children (according to BMI measurement) than excessive body weight. This fact was omitted in the publication.Correct methodology. I propose to expand the research group.

Reviewer 2 Report

First of all, I would like to thank the journal and its editor for the possibility of reviewing this manuscript entitled "Prevalence of Obesity and Associated Risk Factors Among 2 Children and Adolescents in the Eastern Cape Province" that chooses to be published in the IJERPH journal.

The manuscript deals with a current and worrying topic at the same time, obesity, the young population and associated risk factors. To make matters worse, this study takes place in a special area, in Africa, where, as the study says, it is going from a low percentile to being overweight, both being negative for health.
The study is powerful and well written. The introduction, despite being well founded, mentions various countries which are not related to each other. There is talk of Mexico, China, etc. but a relationship, a cohesion, that they are connected, is not mentioned. Also some data could be used in the discussion.
It would be interesting if the introduction ended with a sentence with the objective of the study.
The discussion section, in the first sentence, says the objective, it should not go, since it is already known what the objective of the study is because it has been mentioned at the beginning of the manuscript.
I am missing a section on study limitations, has this study not had limitations?
The conclusions are more than broad, they should be more specific and brief. Despite its length, it is worth noting the good implications for practice and solutions for the future.

Reviewer 3 Report

I am grateful for the opportunity to review the manuscript presented to me. I hope that the comments in the review would be helpful
i I believe the paper is worth considering for publication, however requires
major revision. 

Round 2

Reviewer 1 Report

After the authors' corrections, I accept the paper for publication. I still believe that the small study group is a serious limiting factor.

Author Response

Dear Reviewer

Thank you for the constructive comments and wonderful work. We have taken note. 

The study focused on specific population of the Prospective Urban Rural Epidemiology which (PURE) only focuses on adult population. The sample size for child population among these families from 102 households was limited.

This has, however, been acknowledged under limitations in lines 526-529.

We hope you find this orderly.

Reviewer 3 Report

Thank you for the corrections made. You still have not considered or answered some of the questions. For example, I asked to shorten the conclusions to 3-4 sentences - yes, the content was reduced, but it is still too extensive a section. Additionally, we do not provide citations in Conlusion. I asked for the Cronbah alpha calculation in the methodological part and I did not receive a reply. Additionally, in response to the reviewer, please send 2 attachments: one with an indication of the response to a specific comment with an indication of which part it concerns, and the other with an indication of these changes in the file. 

Author Response

Dear Reviewer 

Thank you for the wonderful work and constructive comments. 

We decided to put comments using this rout and attach the manuscript with track changes for your reference as the submission platform only allowed us one attachment.  We hope that will be acceptable with you.

We have provided them to the Editors. Hope it is fine with you.

We hope you find this orderly. 
